# SCMFMDA: Predicting microRNA-disease associations based on similarity constrained matrix factorization

**Lei Li**[1], **Zhen Gao**[1], **Yu-Tian Wang**[1], **Ming-Wen Zhang**[1], **Jian-Cheng Ni**[1]\*, **Chun-Hou Zheng**[1,2]\*, **Yansen Su**[2]\*

**1** School of Cyber Science and Engineering, Qufu Normal University, Qufu, China, **2** School of Artifial Intelligence, Anhui University, Hefei, China

\* nijch@163.com (J-CN); zhengch99@126.com (C-HZ); suyansen@ahu.edu.cn (Y-SS)

## Abstract

miRNAs belong to small non-coding RNAs that are related to a number of complicated biological processes. Considerable studies have suggested that miRNAs are closely associated with many human diseases. In this study, we proposed a computational model based on Similarity Constrained Matrix Factorization for miRNA-Disease Association Prediction (SCMFMDA). In order to effectively combine different disease and miRNA similarity data, we applied similarity network fusion algorithm to obtain integrated disease similarity (composed of disease functional similarity, disease semantic similarity and disease Gaussian interaction profile kernel similarity) and integrated miRNA similarity (composed of miRNA functional similarity, miRNA sequence similarity and miRNA Gaussian interaction profile kernel similarity). In addition, the $L_2$ regularization terms and similarity constraint terms were added to traditional Nonnegative Matrix Factorization algorithm to predict disease-related miRNAs. SCMFMDA achieved AUCs of 0.9675 and 0.9447 based on global Leave-one-out cross validation and five-fold cross validation, respectively. Furthermore, the case studies on two common human diseases were also implemented to demonstrate the prediction accuracy of SCMFMDA. The out of top 50 predicted miRNAs confirmed by experimental reports that indicated SCMFMDA was effective for prediction of relationship between miRNAs and diseases.

## Author summary

Considerable studies have suggested that miRNAs are closely associated with many human diseases, so predicting potential associations between miRNAs and diseases can contribute to the diagnose and treatment of diseases. Several models of discovering unknown miRNA-diseases associations make the prediction more productive and effective. We proposed SCMFMDA to obtain more accuracy prediction result by applying similarity network fusion to fuse multi-source disease and miRNA information and utilizing similarity constrained matrix factorization to make prediction based on biological information. The global Leave-one-out cross validation and five-fold cross validation were

**Data Availability Statement:** All relevant data are within the manuscript and its Supporting Information files.

**Funding:** This work was supported by the National Natural Science Foundation of China through grants 61873001 (C.Z., Y.W.), U19A2064 (C.Z.), 61872220 (C.Z.) and 11701318 (Y.W.), the Natural Science Foundation of Shandong Province grant ZR2020KC022 (J.N., Y.W., Z.G., L.L) and the Open Project of Anhui Provincial Key Laboratory of Multimodal Cognitive Computation, Anhui University, No. MMC202006 (Y.W., L.L). The funders had no role in study design, data collection and analysis, decision to publish, or preparation of the manuscript.

**Competing interests:** The authors have declared that no competing interests exist.

applied to evaluate our model. Consequently, SCMFMDA could achieve AUCs of 0.9675 and 0.9447 that were obviously higher than previous computational models. Furthermore, we implemented case studies on significant human diseases including colon neoplasms and lung neoplasms, 47 and 46 of top-50 were confirmed by experimental reports. All results proved that SCMFMDA could be regard as an effective way to discover unverified connections of miRNA-disease.

## Introduction

MicroRNAs (miRNAs) are a number of 17-24nt non-coding RNAs, which act a pivotal part in controlling the expression of gene through RNA cleavage or translation repression [1–3]. Lin-4 was the first miRNA inspected in experiment by Lee et al. [4] in 1993. Since that time, a large amount of miRNAs was discovered by researchers in experiments [4,5]. Researchers have sought out generous miRNAs from various of species that included viruses, animals and plants [6]. Because miRNAs regulated the expression of a great quantity of target genes, the total miRNA pathway played a key role in gene expression control [7–9]. miRNAs are bound up with several crucial biological processes, such as cell development, cell differentiation, cell proliferation and so on [10]. Developmental defects can be the result of the dysregulation of miRNAs that also associate with progression of diseases [11]. In the meantime, considerable studies have indicated that miRNAs are connected with a serious of human neoplasms, which include lung neoplasms [12], prostate neoplasms [13] and so on. Hence distinguishing miRNAs associated with diseases can deepen understanding of the genetic causes of complex diseases. Massive connections between miRNAs and diseases have been found by a variety of traditional experiments in the past few years [14,15]. Traditional manual models can infer the connections between miRNA and disease, but which are time-consuming, laborious and high failure rate. Therefore, showing the potential relationship between miRNAs and diseases in need of computational methods with effectiveness and stability, as they can obtain increasing reliable miRNA-disease connections [16].

In the past period of time, a great deal of computation-based algorithms and methods have been applied to predict potential relationship of miRNA-disease [17,18]. For example, Jiang et al. [19] proposed a model that applied the human phenome-microRNAome network to predict potential interactions between miRNAs with similar function and diseases with similar phenotypic. However, the predictive performance of the model was not as decent as expected due to be affected by high false positive and false negative rates existing in the associations between miRNAs and targets. Later, the model WBSMDA [20] introduced the Gaussian interaction profile similarity to enrich similarity information of miRNA and disease. The WBSMDA could also predict potential relationship between new miRNAs and new diseases without any verified correlative information. The collaborative matrix factorization method was applied to predict the relationship of miRNA-diseases in CMFMDA [21], which also could utilize plentiful biological information observe unknown interactions. The model EGBMMDA [22] began to take advantage of decision tree learning to discover novel miRNA-disease interaction by integrating verified miRNA-disease connections, miRNA functional similarity and disease semantic similarity. The informative feature vector was constructed by multi-measures to train the regression tree under the gradient boosting framework. Zhao et al. [23] applied adaptive boosting to observe unverified miRNA-disease association in ABMDA model. And they utilized k-means clustering on negative samples to perform random sampling, which could control the balance between positive samples and negative samples. The

BHCMDA [24] model utilized biased heat conduction (BHC) algorithm to predict unknown connections between miRNAs and diseases though combining miRNA similarity matrix, disease similarity matrix and miRNA-disease association matrix. The probabilistic matrix factorization (PMF) algorithm was used in IMIPMF [25] model to infer potential miRNA-disease interactions. The PMF was widely used in recommender systems, so it could effectively make use of all information to recommend miRNAs which are strongly associated with the disease.

Recently, the methods based on random walk were gradually proposed and more accuracy prediction results were obtained. Chen et al. [26] utilized the random walk with restart algorithm to construct RWRMDA model. Because the prediction performance calculated by global network similarity was better than local network [27,28], RWRMDA employed global network similarity to gain the feasible interactions between miRNAs and diseases. Unfortunately, RWRMDA was inappropriate to the diseases without known associated miRNAs. Shi et al. [29] utilized the function links between human disease genes and miRNA targets to devise a novel model. Random walk algorithm and global network distance measurement were applied to search feasible relationship between miRNAs and diseases. Liu et al. [30] also implemented random walk with restart algorithm in the model to make prediction results to a higher degree. They employed random walk with restart algorithm on a heterogeneous graph established by utilizing disease similarity and miRNA similarity. Luo et al. [31] employed imbalanced bi-random walk method on a heterogeneous network with information of miRNAs and diseases to identify feasible interactions of miRNA-disease. Niu et al. [32] applied random walk with restart algorithm to extract miRNA features from integrated miRNA similarity network in RWBRMDA model. Then these miRNA features were utilized by binary logistic regression algorithm to predict potential miRNA-disease associations.

For the sake of obtaining reliable and accurate predictive performance, machine learning-based methods gradually were utilized to predict unknown miRNA-disease associations. For instance, the model RBMMMDA [33] utilized restricted Boltzmann machine to predict miRNA-disease multi-type associations. The RBMMMDA could gain not only novel associations between miRNAs and diseases, but also corresponding association types. The model PBMDA [34] constructed a heterogeneous graph including different interlinked sub-graphs and further adopted depth-first search algorithm to seek potential miRNA-disease associations. PBMDA could function as a useful calculation tool to accelerate the prediction of miRNA-disease interactions. The model DNRLMF-MDA [35] integrated dynamic neighborhood regularized and logistic matrix factorization to predict potential relationship of miRNA-disease. DNRLMF-MDA applied logistic matrix factorization algorithm to association probability between miRNAs and diseases. Then implementing dynamic neighborhood regularized algorithm to improve predictive performance. Peng et al. [36] proposed the model MDA-CNN for miRNA-disease connection identification. The miRNA-disease interaction features were firstly captured by a three-layer network. Then an auto-encoder was employed to identify obvious miRNA-disease feature combinations. After these feature representations were reduced, the convolutional neural network utilized them to predict the final results. The significant machine learning-based model MLMDA [37] was proposed by Zheng et al. to predict unknown relationship of miRNA-disease. The k-mer sparse matrix was used to extract miRNA sequence information. Then integrating miRNA sequence information, miRNA and disease similarity information to construct feature vectors. The deep auto-encoder neural network (AE) and random forest classifier made full use of feature vectors to calculate the prediction probability. The NCMCMDA [38] model integrated neighborhood constraint with matrix completion algorithm to change the recovery task into an optimization problem. This model applied the fast iterative shrinkage-thresholding algorithm to recover missing interactions between miRNAs and diseases. Zhang et al. [39] proposed the computational model

MSFSP to achieve a more accuracy predictive performance of miRNA-disease interactions. The MSFSP firstly integrated various similarity information of miRNA and disease to construct the similarity of miRNA and disease. Then miRNA and disease similarity matrices and verified miRNA-disease association matrix were utilized to constitute the weighted network of miRNA-disease connections. The final prediction labels were calculated by weighting miRNA and disease space projection scores. Ji et al. [40] proposed SVAEMDA model to infer more disease-related miRNAs, which used miRNA similarity and disease similarity to obtain the representations of miRNA and disease. In addition, the variational autoencoder based predictor was trained to predict unknown interactions of miRNA-disease, which combined verified miRNA-disease interactions with the representations of miRNA and disease to generate the feature vectors of miRNA and disease.

Because there were several limitations in previous models, we presented a novel model based on Similarity Constrained Matrix Factorization for miRNA-Disease Association Prediction (SCMFMDA). In order to obtain plentiful disease similarity data, we applied similarity network fusion algorithm to integrate various disease similarities, which consisted of disease functional similarity, disease semantic similarity and disease Gaussian interaction profile kernel similarity. Similarly, miRNA similarity data was obtained by applying similarity network fusion to integrate miRNA functional similarity, miRNA sequence similarity and miRNA Gaussian interaction profile kernel similarity. In addition, we added $L_2$ regularization terms and similarity constraint terms to standard Nonnegative Matrix Factorization (NMF) method to predict more unknown miRNA-disease associations. To evaluate the effectiveness of SCMFMDA, global Leave-one-out cross validation and five-fold cross validation were carried out on the verified miRNA-disease association data downloaded from HMDD v2.0 [41]. As a result, SCMFMDA achieved AUC values of 0.9675 and 0.9447, respectively. Furthermore, we performed case studies on colon neoplasms and lung neoplasms. Consequently, the miR2Disease [42] and dbDEMC v2.0 [43] databases were utilized to validate results of case studies, which achieved high confirmation ratios. Experimental results showed that SCMFMDA was effective for inferring possible relationship between miRNAs and diseases.

## Materials

### Human miRNA-disease associations

In this study, we downloaded verified human miRNA-disease association information from HMDD v2.0 database, which included 5430 known associations between 383 diseases and 495 miRNAs. For the sake of making calculation convenient, we made an adjacency matrix $A \in R^{nd \times nm}$ to indicate the verified miRNA-disease associations. The $nd$ and $nm$ mean the number of diseases and miRNAs, respectively. We used $a_{ij}$ to represent the $(i,j)$th element of matrix $A$. Specifically, The element $a_{ij}$ is set to 1 if disease $d_i$ is related to miRNA $m_j$; and otherwise, it is set to 0.

### Disease functional similarity

The phenotypically similar diseases tend to associate with similar genes. Therefore, we could calculate disease functional similarity based on the functional information of gene. The log-likelihood score (LLS) represents the probability of a functional linkage between different genes, which can be downloaded from the HumanNet database [44] and be normalized as follows:

$$LLS_n(g_a, g_b) = \frac{LLS(g_a, g_b) - LLS_{min}}{LLS_{max} - LLS_{min}} \tag{1}$$

where $LLS(g_a, g_b)$ denotes the LLS between gene $g_a$ and gene $g_b$, $LLS_{max}$ and $LLS_{min}$ are the maximum LLS and minimum LLS in HumanNet database; $LLS_n(g_a, g_b)$ represents the normalized LLS.

Then, the gene functional similarity score can be calculated by the below equation:

$$FS(g_a, g_b) = \begin{cases} 1 & if\ a = b \\ LLS_n(g_a, g_b) & if\ a \neq b \cap e(a, b) \in S_{HumanNet} \\ 0 & if\ a \neq b \cap e(a, b) \notin S_{HumanNet} \end{cases} \tag{2}$$

where $S_{HumanNet}$ represents the link set that contains whole links between genes in HumanNet database; $e(a,b)$ indicates the link between gene $g_a$ and gene $g_b$.

Furthermore, the functional similarity score between gene $g$ and gene set $G$ is defined as follows:

$$S_G(g) = \max_{g_b \in G} FS(g, g_b) \tag{3}$$

The SIDD [45] can be utilized to obtain disease-gene association data, which are involved in calculating disease functional similarity $SD_1$ by the following equation:

$$SD_1(d_i, d_j) = \frac{\sum_{g_a \in G_i} S_{G_i}(g_a) + \sum_{g_b \in G_j} S_{G_j}(g_b)}{|G_i| - |G_j|} \tag{4}$$

## Disease semantic similarity

On the basis of previous study [46], the medical subject headings (Mesh) descriptors could be implemented to calculate disease semantic similarity. Here, the Directed Acyclic Graph (DAG) could be adopted to indicate the specific relationship of different diseases. Concretely, the $DAG(D) = (D,T(D),E(D))$ represents the DAG of disease $D$, in which $T(D)$ denotes the node set containing $D$ itself and its ancestor nodes, $E(D)$ denotes the relevant edge set including edges from parent nodes to their child nodes directly. Then the semantic value of disease $D$ can be calculated as below:

$$DV1(D) = \sum_{d \in T(D)} D_D1(d) \tag{5}$$

where the semantic contribution of disease $d$ to $D$ can be calculated as follows:

$$D_D1(d) = \begin{cases} 1 & if\ d = D \\ \max\{\Delta * D_D1(d') | d' \epsilon children\ of\ d\} & if\ d \neq D \end{cases} \tag{6}$$

here, $\Delta$ is the semantic contribution factor that is set to 0.5 based on previous literature [47].

On the basis of assumption that various diseases tend to be regarded as similar diseases if the large parts of their DAGs are same. Therefore, the semantic similarity $DS_1(d_i, d_j)$ between disease $d_i$ and disease $d_j$ can be defined as follows:

$$DS_1(d_i, d_j) = \frac{\sum_{t \in T(d_i) \cap T(d_j)} (D_{d_i}1(t) + D_{d_j}1(t))}{DV1(d_i) + DV1(d_j)} \tag{7}$$

Based on the previous study [48], diseases appear in less DAGs may be more specific, these diseases ought to gain a higher semantic contribution in DAGs. Therefore, different diseases located in the same layer of one DAG, which may obtain the different contribution value. Specifically, the semantic contribution of disease $d$ to $D$ can be calculated in different way as

below:

$$D_D2(d) = -\log\left(\frac{\text{the number of DAGs including } d}{\text{the number of diseases}}\right) \tag{8}$$

Correspondingly, the semantic score of disease $D$ and semantic similarity $DS_2(d_i, d_j)$ between disease $d_i$ and disease $d_j$ can be calculated as follows:

$$DV2(D) = \sum_{d \in T(D)} D_D2(d) \tag{9}$$

$$DS_2(d_i, d_j) = \frac{\sum_{t \in T(d_i) \cap T(d_j)} (D_{d_i}2(t) + D_{d_j}2(t))}{DV2(d_i) + DV2(d_j)} \tag{10}$$

Finally, we integrated $DS_1$ and $DS_2$ to calculate final disease semantic similarity $SD_2(d_i, d_j)$ between disease $d_i$ and disease $d_j$ in following equation:

$$SD_2(d_i, d_j) = \frac{DS_1(d_i, d_j) + DS_2(d_i, d_j)}{2} \tag{11}$$

## miRNA functional similarity

Based on the calculation method of miRNA functional similarity [49,50], assuming that functionally similar miRNAs tend to be linked with phenotypically similar diseases and vice versa. We downloaded miRNA functional similarity data from http://www.cuilab.cn/files/images/cuilab/misim.zip. Here, we constructed the matrix $SM_1$ with $nm$ rows and $nm$ columns for storing the corresponding information. The element $SM_1(m_i, m_j)$ represents the relevant functional similarity score between miRNA $m_i$ and miRNA $m_j$.

## miRNA sequence similarity

We utilized the Needleman-Wunsch Algorithm to calculate miRNA sequence similarity, and corresponding miRNA sequence information can be obtained from miRBase database [51]. Be similar to miRNA functional similarity, we also constructed a matrix $SM_2 \in R^{nm \times nm}$ to store sequence similarity information, where $SM_2(m_i, m_j)$ was the relevant sequence similarity score between miRNA $m_i$ and miRNA $m_j$.

## Gaussian interaction profile kernel similarity for diseases and miRNAs

On the basis of previous study [49,50], because miRNAs with similar function are likely to be linked with diseases with similar phenotypes, the Gaussian interaction profile (GIP) kernel similarity can be calculated and applied to stand for the miRNA similarity and disease similarity. Concretely, the binary vector $K(d_i)$ is constructed to indicate the interaction profile of disease $d_i$ in accordance with whether $d_i$ possesses known association with each miRNA or not. Here, the GIP kernel similarity $SD_3(d_i, d_j)$ between disease $d_i$ and disease $d_j$ can be calculated as below equations:

$$SD_3(d_i, d_j) = \exp(-\rho_d ||K(d_i) - K(d_j)||^2) \tag{12}$$

$$\rho_d = \rho'_d / (\frac{1}{nd} \sum_{i=1}^{nd} ||K(d_i)||^2) \tag{13}$$

In the same light, the GIP kernel similarity $SM_3(m_i, m_j)$ between miRNA $m_i$ and miRNA $m_j$ can be calculated by the following formulas:

$$SM_3(m_i, m_j) = \exp(-\rho_m ||K(m_i) - K(m_j)||^2) \tag{14}$$

$$\rho_m = \rho'_m / (\frac{1}{nm} \sum_{i=1}^{nm} ||K(m_i)||^2) \tag{15}$$

where the binary vector $K(m_i)$ indicates the interaction profile of miRNA $m_i$ in accordance with whether $m_i$ has known association with each disease or not, the parameter $\rho_m$ is utilized to control kernel bandwidth.

## Methods

### Overview

The SCMFMDA includes two major parts: similarity network fusion is applied to obtain integrated disease similarity and integrated miRNA similarity; known miRNA-diseases associations and integrated similarities are adopted in similarity constrained matrix factorization to infer unknown associations of miRNA-disease. The specific flow chart of SCMFMDA is shown in Fig 1.

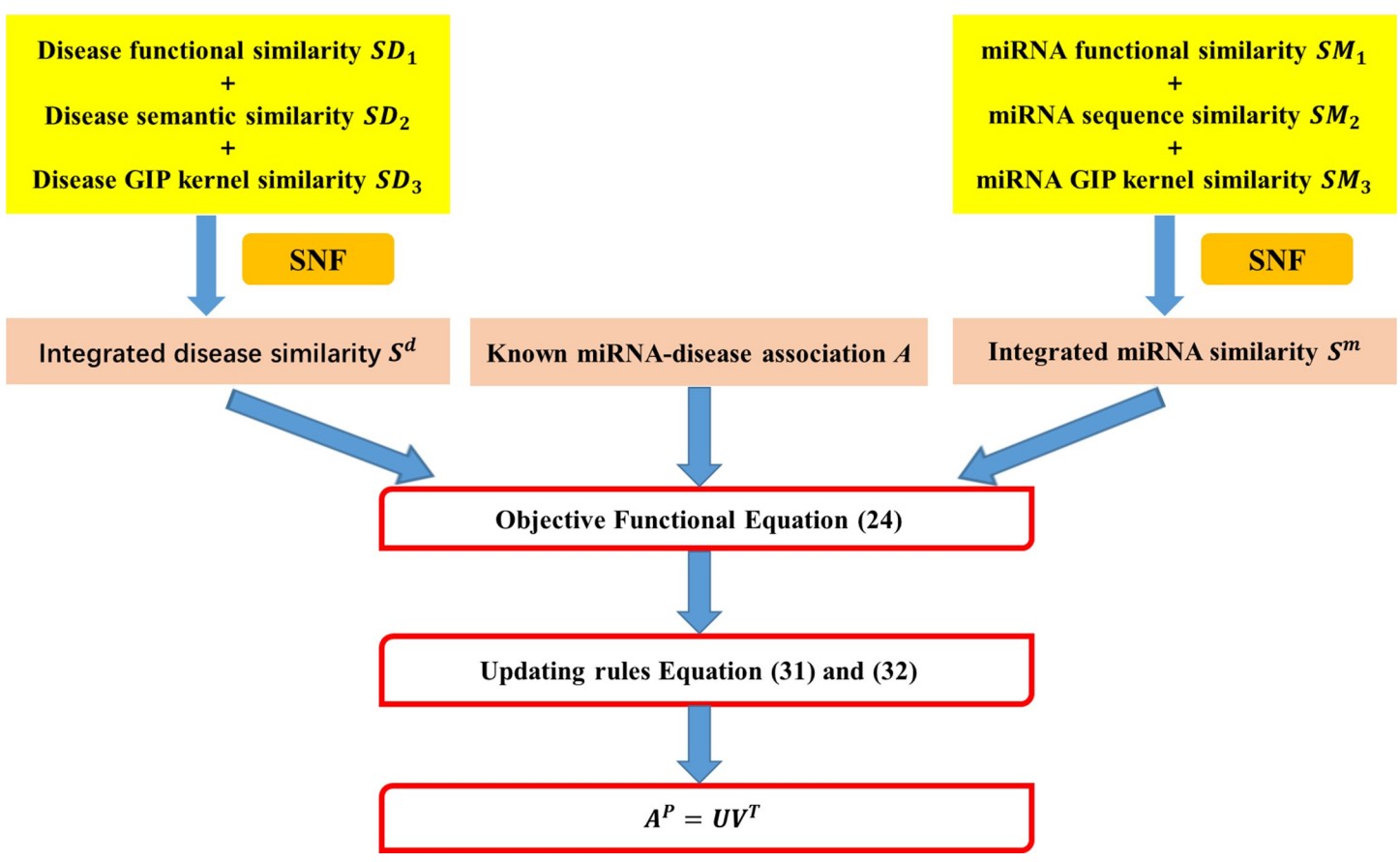

**Fig 1. Flow chart of SCMFMDA.**

## Integrating similarity for diseases and miRNAs

The similarity between two diseases can use disease functional similarity, disease semantic similarity and disease GIP kernel similarity to represent. Similarly, miRNA functional similarity, miRNA sequence similarity and miRNA GIP kernel similarity can be utilized to indicate similarity between different miRNAs. Here, the similarity network fusion (SNF) [52] method is applied to integrate various similarities for disease and miRNA. According to previous study, the process of SNF can be expressed as iterative update of similarity matrices. The main steps of utilizing SNF to integrate different disease similarities $SD_n$, $n = 1,2,3$ are introduced as follows.

In the first step, we calculated normalized weight matrix $P_n$ of each similarity network as follows:

$$P_n(d_i, d_j) = \begin{cases} \dfrac{SD_n(d_i, d_j)}{2\sum_{k \neq i} SD_n(d_i, d_k)} & j \neq i \\ \dfrac{1}{2} & j = i \end{cases} \tag{16}$$

In the second step, we utilized $k$ nearest neighbor (KNN) algorithm to measure the local relationship of each similarity network. The specific process to obtain corresponding matrix $K_n$ is displayed as follows:

$$K_n(d_i, d_j) = \begin{cases} \dfrac{SD_n(d_i, d_j)}{\sum_{k \in N_i} SD_n(d_i, d_k)} & j \in N_i \\ 0 & otherwise \end{cases} \tag{17}$$

where the $N_i$ indicates the number of neighbors in the disease.

In the third step, we applied SNF to integrate normalized weight matrix $P_n$ and local relationship matrix $K_n$ as follows:

$$P_n = K_n \left( \frac{\sum_{t \neq n} P_t}{m - 1} \right) (K_n)^T \quad n = 1, 2, \ldots, m \tag{18}$$

Because we had three different disease similarity networks (disease functional similarity, disease semantic similarity and disease GIP kernel similarity), the m was equal to 3. After iterative update, the ultimate disease similarity matrix $S^d$ could be obtained as follows:

$$S^d = \frac{1}{3} \sum_{n=1}^{3} P_n \tag{19}$$

Similarly, we could apply SNF algorithm to obtain final miRNA similarity matrix $S^m$.

## Similarity constrained matrix factorization

After obtaining processed disease similarity and miRNA similarity, similarity constrained matrix factorization method is adopted to observe more unknown interactions of miRNA-disease, and Fig 2 shows concrete details of it. The SCMFMDA factorized the matrix $A \in R^{nd \times nm}$ into $U \in R^{nd \times \gamma}$ and $V \in R^{nm \times \gamma}$, where $\gamma$ denoted the dimension of disease feature and miRNA feature in the low-rank spaces. To be specific, the association of miRNA-disease roughly equal to the inner product between the disease feature vector and the miRNA feature vector: $a_{ij} \approx u_i v_j^T$, where $u_i$ and $v_j$ represent the $i$th row of U and the $j$th row of V, respectively. The

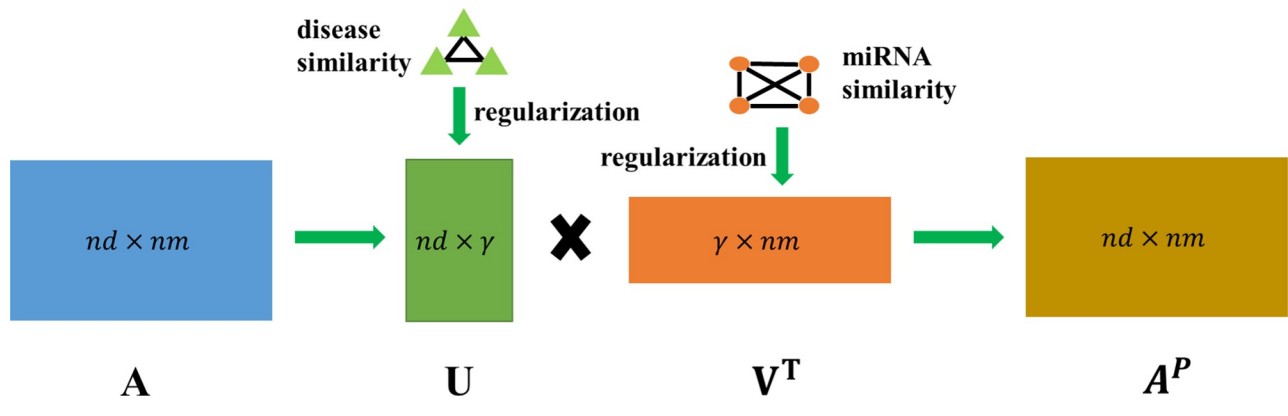

**Fig 2. The details of similarity constrained matrix factorization.**

corresponding objective function is shown as follows:

$$\min \frac{1}{2}\sum_{ij}(a_{ij} - u_i v_j^T)^2 \tag{20}$$

Then, the $L_2$ regularization terms of $u_i$ and $v_j$ are added to the Eq (20) for solving overfitting problem.

$$\min \frac{1}{2}\sum_{ij}(a_{ij} - u_i v_j^T)^2 + \frac{\sigma}{2}\sum_{i}||u_i||^2 + \frac{\sigma}{2}\sum_{j}||v_j||^2 \tag{21}$$

where $\sigma$ is the regularization parameter for $u_i$ and $v_j$.

On the basis of previous study [53,54], the geometric properties of data points may be kept when they are mapped from high-rank space into low-rank space. Disease similarity $S^d$ and miRNA similarity $S^m$ can indicate geometric structure of data points, so we present similarity constraint terms $S_U$ and $S_V$ as follows:

$$S_U = \frac{1}{2}\sum_{ij}||u_i - u_j||^2 S_{ij}^d \tag{22}$$

$$S_v = \frac{1}{2}\sum_{ij}||v_i - v_j||^2 S_{ij}^m \tag{23}$$

where $S_{ij}^d$ represents the similarity between disease $d_i$ and disease $d_j$, $S_{ij}^m$ denotes the similarity between miRNA $m_i$ and miRNA $m_j$, respectively. Considering the similarity degree between two data points is up to the distance of them, so $S_U$ will incur a heavy penalty if the distance of $d_i$ and $d_j$ are close in disease feature space. Therefore, we could keep the geometric structure of disease data points by minimizing $S_U$, which would cause that disease $d_i$ and disease $d_j$ were mapped closely in low dimensional space. For miRNA, it is the same situation. Hence, the objective function of SCMFMDA are proposed by adding $S_U$ and $S_V$ to Eq (21) as follows:

$$\min_{U,V} L = \frac{1}{2}\sum_{ij}(a_{ij} - u_i v_j^T)^2 + \frac{\sigma}{2}\sum_{i}||u_i||^2 + \frac{\sigma}{2}\sum_{j}||v_j||^2 + \frac{\varepsilon}{2}\sum_{ij}||u_i - u_j||^2 S_{ij}^d + \frac{\varepsilon}{2}\sum_{ij}||v_i - v_j||^2 S_{ij}^m \tag{24}$$

where $\varepsilon$ is regarded as hyper parameter which can availably control the smoothness of similarity consistency.

## Optimization algorithm

In this section, we proposed an efficacious optimization algorithm to calculate the objective function of SCMFMDA. First, the partial derivatives of $L$ in regard to $u_i$ and $v_j$ are calculated as follows:

$$\nabla_{u_i}L = \sum_j (u_i v_j^T - a_{ij})v_j + \sigma u_i + \varepsilon\left(\sum_j(u_i - u_j)S_{ij}^d - \sum_j(u_j - u_i)S_{ji}^d\right)$$
$$= u_i\left(V^TV + \sigma I + \varepsilon(\sum_j S_{ij}^d + \sum_j S_{ji}^d)I\right) - A(i,:)V - \varepsilon\sum_j(S_{ij}^d + S_{ji}^d)u_j \quad (25)$$

where $A(i,:)$ denotes the $i$th row of matrix $A$.

$$\nabla_{v_j}L = \sum_i (v_j u_i^T - a_{ij})u_i + \sigma v_j + \varepsilon\left(\sum_i(v_j - v_i)S_{ji}^m - \sum_i(v_i - v_j)S_{ij}^m\right)$$
$$= v_j\left(U^TU + \sigma I + \varepsilon(\sum_i S_{ij}^m + \sum_i S_{ji}^m)I\right) - A(:,j)^TU - \varepsilon\sum_i(S_{ij}^m + S_{ji}^m)v_i \quad (26)$$

where $A(:,j)$ denotes the $j$th column of matrix $A$.

Then, the second derivatives of $L$ in regard to $u_i$ and $v_j$ are calculated by the below equations:

$$\nabla_{u_i}^2 L = V^TV + \sigma I + \varepsilon\left(\sum_j S_{ij}^d + \sum_j S_{ji}^d\right)I \quad (27)$$

$$\nabla_{v_j}^2 L = U^TU + \sigma I + \varepsilon\left(\sum_i S_{ij}^m + \sum_i S_{ji}^m\right)I \quad (28)$$

According to Newton's method, $u_i$ and $v_j$ can be executed iterative update as follows:

$$u_i \leftarrow u_i - \nabla_{u_i}L(\nabla_{u_i}^2 L)^{-1} \quad (29)$$

$$v_j \leftarrow v_j - \nabla_{v_j}L(\nabla_{v_j}^2 L)^{-1} \quad (30)$$

Hence, $u_i$ and $v_j$ can be updated by the following formulas:

$$u_i = \left(A(i,:)V + \varepsilon\sum_j(S_{ij}^d + S_{ji}^d)u_j\right)\left(V^TV + \sigma I + \varepsilon\left(\sum_j S_{ij}^d + \sum_j S_{ji}^d\right)I\right)^{-1} \quad (31)$$

$$v_j = \left(A(:,j)^TU + \varepsilon\sum_i(S_{ij}^m + S_{ji}^m)v_i\right)\left(U^TU + \sigma I + \varepsilon\left(\sum_i S_{ij}^m + \sum_i S_{ji}^m\right)I\right)^{-1} \quad (32)$$

When the convergence condition is met, the update of $u_i$ and $v_j$ will stop. The prediction matrix can be obtained by updated $u_i$ and $v_j$.

$$A^P = UV^T \quad (33)$$

The value of $A_{ij}^P$ denotes the association probability between disease $d_i$ and miRNA $m_j$. The more likely the association is, if the score is higher.

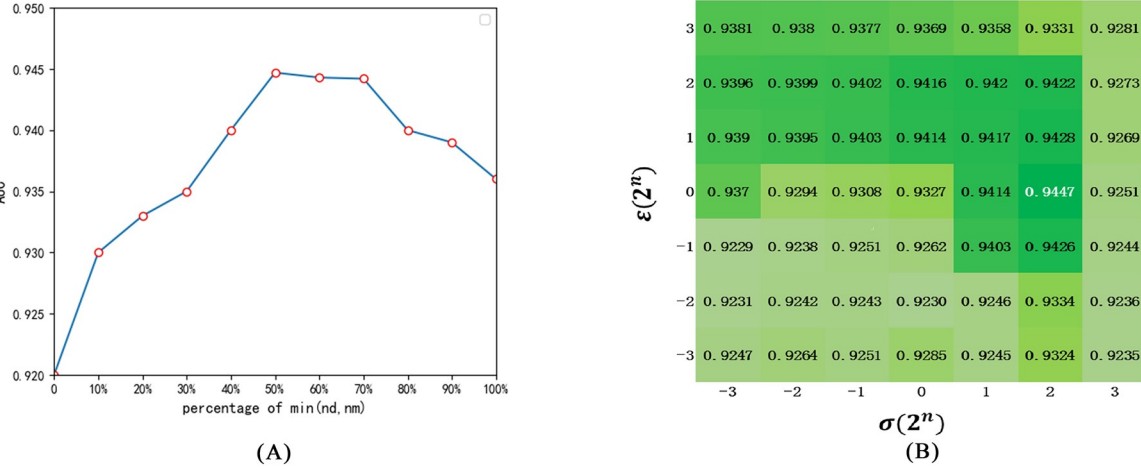

**Fig 3.** The influence of parameters on SCMFMDA: (A) the influence of $\gamma$; (B) the influence of $\sigma$ and $\varepsilon$.

## Results

### Parameters optimization

In this section, parameters $\gamma$, $\sigma$ and $\varepsilon$ are quantitatively analyzed to research their effect on the prediction performance. $\gamma$ represents the dimension of diseases and miRNAs in low-rank spaces, and $\gamma <$ min ($nd$, $nm$) that can be considered as the percentage of min ($nd$, $nm$). Parameters $\sigma$ and $\varepsilon$ denote the regularization parameters. The AUC value of 5-CV is applied to evaluate influence of the choice of parameters on the performance of model. And after generous test experiments were conducted, we could get the conclusion that the value of $\gamma$ would affect the experiment individually. For this reason, we fixed $\sigma$ and $\varepsilon$ in a suitable combination to test the most suitable value of $\gamma \in \{0, 10\%, \ldots, 1\}$ in SCMFMDA. In order to ensure the correctness of the test, $\sigma$ and $\varepsilon$ are fixed in different combination. From Fig 3A, we could see that SCMFMDA obtained the best performance when $\gamma = 50\%$. In addition, the $\gamma = 50\%$ is fixed so that the effect of regularization parameters $\sigma$ and $\varepsilon$ can be clearly evaluated. We utilized all combinations of $\sigma \in \{2^{-3}, 2^{-2}, \ldots, 2^3\}$ and $\varepsilon \in \{2^{-3}, 2^{-2}, \ldots, 2^3\}$ to construct SCMFMDA. From Fig 3B, we could discover that SCMFMDA acquired best AUC value of 0.9447 when $\sigma = 2^2$ and $\varepsilon = 2^0$. In summary, $\gamma$, $\sigma$ and $\varepsilon$ are set to 50%, $2^2$ and $2^0$ in our model, respectively.

### Model comparison

In order to evaluate the prediction ability of SCMFMDA, we compared several previous computational methods that were proposed to predict unknown miRNA-disease associations. We applied same dataset (HMDD v2.0 database) to train these methods so that comparison results could be considered as fairness. The specific information of these methods are shown as follows.

- MSCHLMDA [55] is a multi-similarity based combinative hypergraph learning model (published in 2020).

- ICFMDA [56] is an improved collaborative filtering-based computational model (published in 2018).

- SACMDA [57] is short acyclic connections-based computational model (published in 2018).

- GRNMF [58] is a graph regularized non-negative matrix factorization-based model (published in 2018).

- $GRL_{2,1}-NMF$ [59] is a graph Laplacian regularized $L_{2,1}$-nonnegative matrix factorization-based computational model (published in 2020).

- NPCMF [60] is a nearest profile-based collaborative matrix factorization model (published in 2019).

- KBMFMDA [61] is a kernelized Bayesian matrix factorization-based computational model (published in 2020).

Based on the HMDD v2.0 database that included 5430 verified associations and 184155 unverified associations between 383 diseases and 495 miRNAs, global Leave-one-out cross validation (global LOOCV) and five-fold cross validation (5-CV) were implemented to evaluate the prediction performance of these methods. In the framework of global LOOCV, the test set was held by each verified association of miRNA-disease in turn, the training set was composed of other verified associations. The whole unknown miRNA-disease associations were considered as candidate samples. Similarly, in the framework of 5-CV, the whole verified miRNA-disease associations were divided into five parts in a random way, where test set was held by one part in turn, training set consisted of other four parts in turn. The whole unknown miRNA-disease associations were considered as candidate samples. In addition, by either the global LOOCV or the 5-CV, we applied SCMFMDA to obtain all predicted association scores so that the ranking of test set relative to candidate samples could be calculated. When the ranking of all test sample were higher than the certain threshold, SCMFMDA was regarded as a valid model. Then we could utilize the Receiver operating characteristics (ROC) curve that was obtained by plotting the true positive rate (TPR) against the false positive rate (FPR) to effectively evaluate the performance of SCMFMDA. We could calculate the area under the ROC curve (AUC) of SCMFMDA whose value was between 0 and 1. Similarly, we could obtain AUCs of other computational methods by utilizing the information of HMDD v2.0 database.

In this work, when global LOOCV method was conducted, SCMFMDA, MSCHLMDA, ICFMDA and SACMDA acquired average AUC values of 0.9675, 0.9287, 0.9072 and 0.8777, respectively (Fig 4). For the purpose of reducing potential deviations resulted in random sample segmentations, we applied 100 times repeated segmentations to verified associations of miRNA-disease in 5-CV method, and the average AUC values of SCMFMDA, MSCHLMDA, ICFMDA and SACMDA reached 0.9447, 0.9263, 0.9046, and 0.8773, respectively (Fig 5). Obviously, the prediction performance of SCMFMDA was better than other methods.

In order to further reflect the performance of the SCMFMDA, it is also compared with other state-of-the-art matrix factorization-based methods that include GRNMF, $GRL_{2,1}-NMF$, NPCMF, KBMFMDA. The 5-CV results of all model are demonstrated in Table 1, clearly SCMFMDA possesses the best AUC. The advantages of SCMFMDA than other matrix factorization-based models are as follows: first, the biological similarity data that are utilized in SCMFMDA obviously more than other models; second, SCMFMDA utilizes SNF instead of traditional linear combination method to integrate various similarity data, which greatly guarantee the completeness and effectiveness of experiment data; third, the $L_2$ regularization and similarity constraint terms are added to the NMF objective function, which benefit to correctly discover more unknown miRNA-disease connections.

## Case studies

For the purpose of demonstrating the effectiveness and accuracy of SCMFMDA, we applied an evaluation experiment in this section. We implemented two types of human diseases, i.e.,

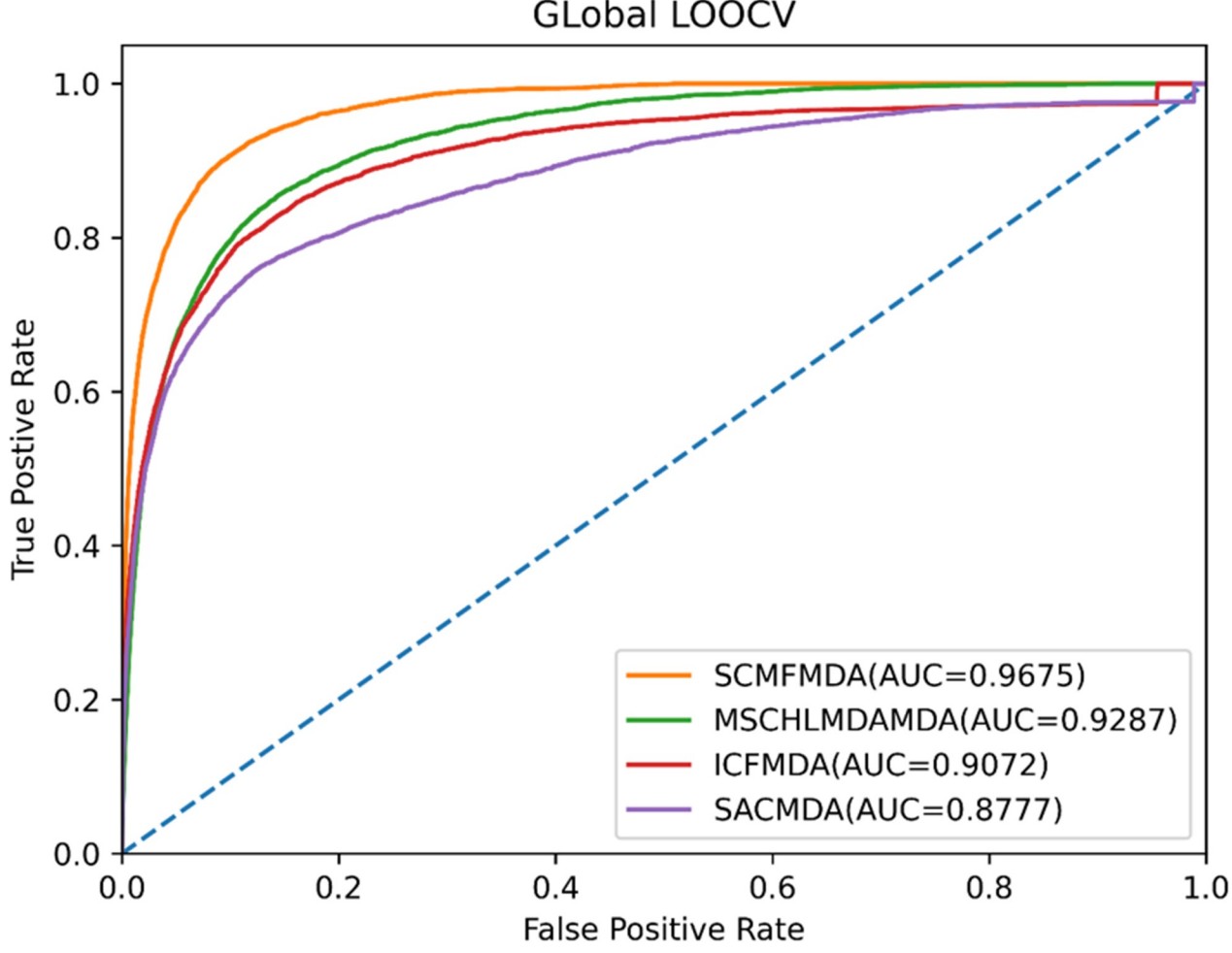

**Fig 4. AUC of global LOOCV compared with those of MSCHLMDA, ICFMDA and SACMDA.**

colon neoplasms and lung neoplasms to validate the expression of our method. There is no doubt that these diseases do great harm to human health. Colon neoplasms belongs to malignancy in the field of Medicine, which has been confirmed to associate with several miRNAs [62,63]. Lung neoplasms is one of the most dangerous malignancies with the fastest increase in morbidity and mortality [12]. A growing number of evidence indicates that lung neoplasms and a few of miRNAs have close relationship. For a specific disease, verified associations of whole diseases in HMDD v2.0 database are considered as training samples, unverified associations with the specific disease in HMDD v2.0 database are treated a candidate samples. By training this model, we could rank predicted association score of the candidate samples and then the top 50 candidate associations with the specific disease are selected. In addition, we utilized two types of databases that were miR2disease and dbDEMC v2.0 to check out miRNAs that have been ranked. Moreover, Tables 2 and 3 indicated prediction results obtained via SCMFMDA, respectively. The 94% and 92% of top 50 miRNAs that inferred by our model, which were individually confirmed to associate with colon neoplasms and lung neoplasms according to the miR2Disease and dbDEMC v2.0 databases. Only 3 and 4 of top 50 predicted miRNAs that are related colon neoplasm and lung neoplasms could not find clues in the databases.

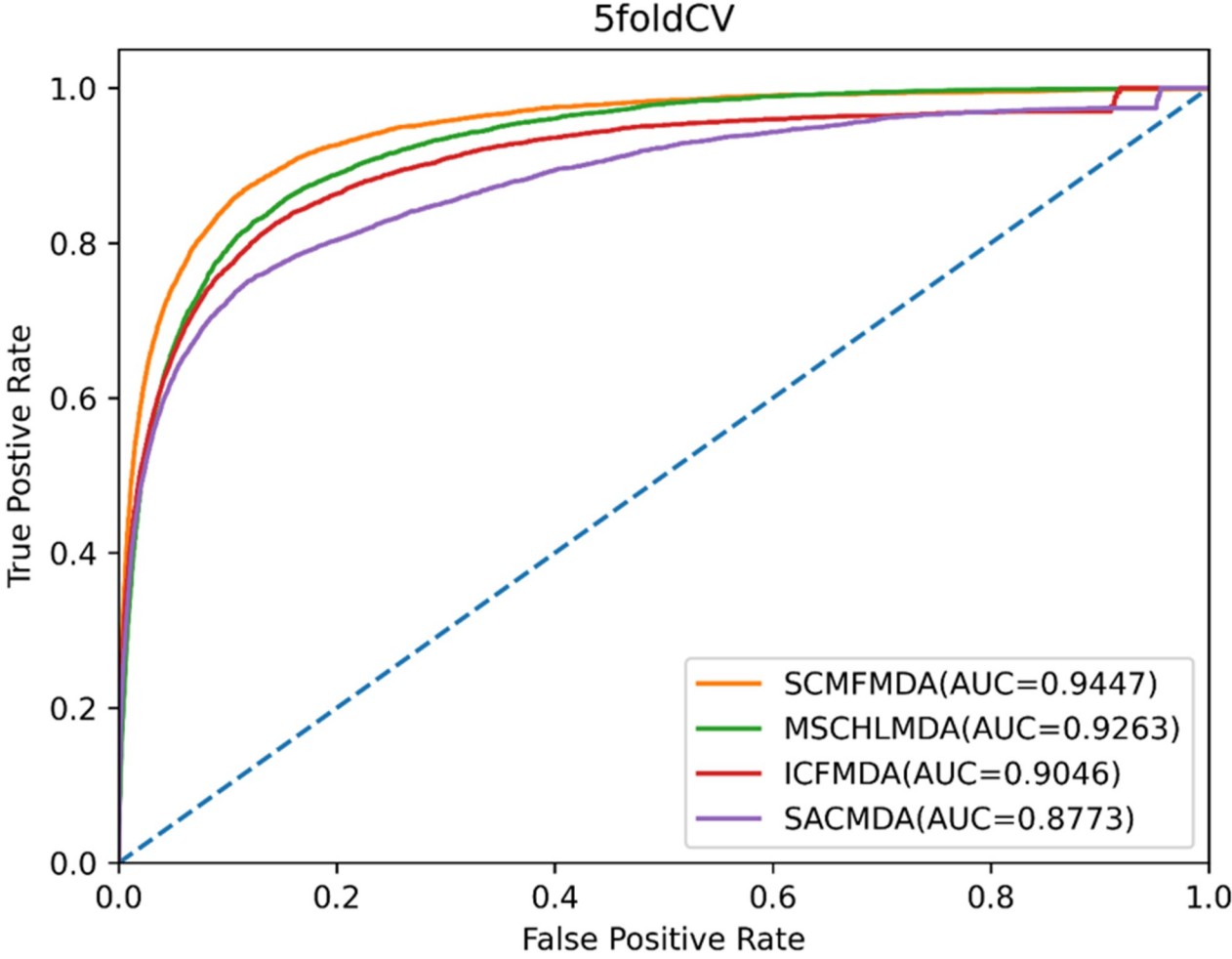

**Fig 5. AUC of 5-CV compared with those of MSCHLMDA, ICFMDA and SACMDA.**

## Discussion and conclusion

In this paper, we introduced a new model named SCMFMDA that used similarity constrained matrix factorization algorithm to predict possible associations of miRNA-disease. In order to obtain plenty of disease similarity data and miRNA similarity data, similarity network fusion algorithm is used to integrate various disease and miRNA biological information, respectively. In addition, $L_2$ regularization terms and similarity constraint terms are added to the standard

**Table 1. Comparisons between SCMFMDA and other MF-based models.**

| Computational models | AUC of 5-CV |
|:---:|:---:|
| GRNMF | 0.869 |
| $GRL_{2,1}-NMF$ | 0.9276 |
| NPCMF | 0.9429 |
| KBMFMDA | 0.9008 |
| SCMFMDA | 0.9447 |

**Table 2. The top 50 potential miRNAs associated with colon neoplasms.**

| miRNA | Evidence | miRNA | Evidence |
|---|---|---|---|
| hsa-mir-21 | a; b | hsa-mir-30a | a; b |
| hsa-mir-20a | a; b | hsa-mir-10b | a; b |
| hsa-mir-143 | a; b | hsa-mir-181b | a; b |
| hsa-mir-155 | a; b | hsa-mir-106b | a |
| hsa-mir-18a | b | hsa-mir-203 | a; b |
| hsa-mir-92a | b | hsa-mir-9 | a; b |
| hsa-mir-34a | a; b | hsa-mir-34c | a |
| hsa-mir-19b | a; b | hsa-mir-196a | a; b |
| hsa-mir-19a | a; b | hsa-mir-183 | a; b |
| hsa-mir-125b | a; b | hsa-mir-142 | unconfirmed |
| hsa-mir-146a | b | hsa-mir-24 | a; b |
| hsa-mir-16 | unconfirmed | hsa-mir-222 | b |
| hsa-mir-200c | a | hsa-mir-133b | a; b |
| hsa-mir-223 | a; b | hsa-mir-34b | a; b |
| hsa-mir-200b | b | hsa-mir-224 | a; b |
| hsa-mir-221 | a; b | hsa-mir-93 | a; b |
| hsa-mir-182 | a; b | hsa-mir-29a | a; b |
| hsa-mir-31 | a; b | hsa-mir-29b | a; b |
| hsa-mir-200a | b | hsa-mir-146b | b |
| hsa-let-7a | a; b | hsa-mir-27a | a; b |
| hsa-mir-205 | b | hsa-mir-210 | b |
| hsa-mir-101 | b | hsa-mir-141 | a; b |
| hsa-mir-218 | b | hsa-mir-148a | b |
| hsa-mir-15a | b | hsa-mir-486 | b |
| hsa-mir-181a | a; b | hsa-mir-199a | unconfirmed |

a: miR2Disease database; b: dbDEMC v2.0 database

NMF for predicting more unobserved miRNA-disease associations. In the frameworks of global LOOCV and 5-CV, the AUCs of SCMFMDA severally achieved 0.9675 and 0.9447 that indicated the performance of our model had a significant improvement relative to previous models. Furthermore, the predicted miRNAs that related to colon neoplasms and lung neoplasms were confirmed by the experiment literatures, so the prediction results of our model were proved to be reliable.

What should be denoted is that the following factors may contribute to the reliable performance of SCMFMDA. First, similarity network fusion algorithm was applied to integrate different disease and miRNA similarities, which can ensure the richness of biological data in the experiment. Then, the function of $L_2$ regularization terms is avoiding overfitting problem. Moreover, the similarity constraint terms consist of disease feature-based similarity and miRNA feature-based similarity, which can generate robustness to the data richness.

However, several limitations may influence the performance of SCMFMDA. First, the model is applicable to the diseases and miRNAs must appear in the selected dataset, but can't make predictions for other diseases and miRNAs. In addition, for some important parameters in SCMFMDA, we hadn't appropriate way to select the most suitable parameters expect carrying out all combinations. Therefore, we should continuously optimize our model to improve its performance in later days.

**Table 3. The top 50 potential miRNAs associated with lung neoplasms.**

| miRNA | Evidence | miRNA | Evidence |
|---|---|---|---|
| hsa-mir-16 | a; b | hsa-mir-378a | unconfirmed |
| hsa-mir-195 | a; b | hsa-mir-92b | b |
| hsa-mir-141 | a; b | hsa-mir-342 | b |
| hsa-mir-106b | B | hsa-mir-367 | b |
| hsa-mir-15a | B | hsa-mir-23b | b |
| hsa-mir-429 | a; b | hsa-mir-139 | a; b |
| hsa-mir-296 | unconfirmed | hsa-mir-373 | b |
| hsa-mir-99a | a; b | hsa-mir-452 | b |
| hsa-mir-122 | B | hsa-mir-148b | b |
| hsa-mir-130a | a; b | hsa-mir-339 | a; b |
| hsa-mir-151a | unconfirmed | hsa-mir-302c | b |
| hsa-mir-625 | B | hsa-mir-302d | b |
| hsa-mir-193b | B | hsa-mir-423 | a |
| hsa-mir-152 | B | hsa-mir-208a | unconfirmed |
| hsa-mir-20b | B | hsa-mir-328 | b |
| hsa-mir-15b | B | hsa-mir-708 | b |
| hsa-mir-451a | B | hsa-mir-211 | b |
| hsa-mir-194 | B | hsa-mir-181d | b |
| hsa-mir-196b | B | hsa-mir-215 | b |
| hsa-mir-302b | B | hsa-mir-302a | b |
| hsa-mir-129 | B | hsa-mir-28 | b |
| hsa-mir-204 | a; b | hsa-mir-153 | b |
| hsa-mir-149 | B | hsa-mir-130b | b |
| hsa-mir-10a | B | hsa-mir-345 | a; b |
| hsa-mir-320a | B | hsa-mir-144 | b |

a: miR2Disease database; b: dbDEMC v2.0 database

## Supporting information

**S1 Table. Known human miRNA-disease associations obtained from HMDD v2.0 database.**
(XLSX)

**S2 Table. Names of 383 diseases involved in known human miRNA-disease associations obtained from HMDD v2.0 database.**
(XLSX)

**S3 Table. Names of 495 miRNAs involved in known human miRNA-disease associations obtained from HMDD v2.0 database.**
(XLSX)

**S4 Table. The constructed disease functional similarity score matrix.**
(XLSX)

**S5 Table. The constructed disease semantic similarity score matrix.**
(XLSX)

**S6 Table. The constructed miRNA functional similarity score matrix.**
(XLSX)

**S7 Table. The constructed miRNA sequence similarity score matrix.**
(XLSX)

## Author Contributions

**Conceptualization:** Lei Li.

**Data curation:** Lei Li.

**Formal analysis:** Lei Li, Zhen Gao, Ming-Wen Zhang.

**Investigation:** Lei Li, Zhen Gao, Ming-Wen Zhang.

**Methodology:** Yu-Tian Wang.

**Project administration:** Yu-Tian Wang, Jian-Cheng Ni, Chun-Hou Zheng.

**Resources:** Jian-Cheng Ni, Chun-Hou Zheng, Yansen Su.

**Software:** Lei Li, Zhen Gao.

**Supervision:** Jian-Cheng Ni, Chun-Hou Zheng.

**Validation:** Zhen Gao, Yu-Tian Wang, Jian-Cheng Ni, Chun-Hou Zheng, Yansen Su.

**Writing – original draft:** Lei Li.

**Writing – review & editing:** Lei Li, Yansen Su.

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
