## [Decision Letter · Decision Letter 0]

29 Apr 2021

Dear Mr Li,

Thank you very much for submitting your manuscript "SCMFMDA: Predicting microRNA-disease Associations based on Similarity Constrained Matrix Factorization" for consideration at PLOS Computational Biology.

As with all papers reviewed by the journal, your manuscript was reviewed by members of the editorial board and by several independent reviewers. In light of the reviews (below this email), we would like to invite the resubmission of a significantly-revised version that takes into account the reviewers' comments.

Please carefully revise your paper according to the reviews.

We cannot make any decision about publication until we have seen the revised manuscript and your response to the reviewers' comments. Your revised manuscript is also likely to be sent to reviewers for further evaluation.

Sincerely,

Quan Zou

Guest Editor

PLOS Computational Biology

Ilya Ioshikhes

Deputy Editor

PLOS Computational Biology

Please carefully revise your paper according to the reviews.

Reviewer's Responses to Questions

**Comments to the Authors:**

Reviewer #1: In this study, Li et al. proposed a computational model to predict miRNA-disease associations based on similarity constrained matrix factorization. Their method, SCMFMDA, achieved better performance than existing ones and accurately predict relations between miRNAs and diseases in the following case studies. The model seems valid and the results they show are promising. However, following concerns should be addressed before considering for publication.

- There are three major issues for model validation: First, the authors should provide more details about the dataset they used for validation. They only mentioned: “Based on the verified association of miRNA-disease in HMDD V2.0 database”, which is not enough for the readers to fully understand the dataset. For examples, how many verified associations included in the dataset? How many unknown cases in the dataset? Do they perform any filtering to the dataset? Second, the authors should summarize the existing methods they compete and show the information including the publication time, the mathematic model they use, the dataset they used for training etc. in a table. The authors should also provide the details about how they compare these methods. To my understanding, it’s better to compare different methods in an independent testing set instead of on their own training set. Different methods can be trained on different datasets, it’s unclear how the authors compare these methods using cross validation. Third, the authors should provide the dataset they use as well as the source codes to execute their and others’ models, so that others can reproduce their results to confirm the correctness.

- As for parameter optimization, I’m not sure why they first fix two to optimize the third and then fix the third to optimize other two. And they don’t mention the exact values of two parameters they use to optimize the third. Just randomly pick some values? A more straightforward way is to test all the 490 possible combinations of three parameters and select the combination with best performance. Is this strategy very time-consuming? Figure 5a is not very professional, the dots representing the exact values should be plotted on the figure with a more smoothed curve to show the trend. The first two values on the figure is weird, why there is no value for 0 but two values for 10%? A color bar should be added for the heatmap in figure 5b. They grey line surround the heatmap can be removed.

- For case studies, a more detailed statistics of the performance can be shown in addition to three tables. For examples, for each disease, how many percentages of miRNAs in the top 50 can find evidences in HMDD v3.2 or dbDEMC v2.0 or both? How many cannot find clues in the database. Also, what’ s the differences between HMDD v3.2 and v2.0, which is used for model training? Are there any overlaps? if so, the authors should exclude those involved in the model training.

- The organization of the “Results” part is not quite reasonable. Logically, the parameter optimization should be in front of model comparison. Model comparison can be separated to non-ML methods and ML methods, but I would suggest putting them together. The last part can be the case studies to further confirm the prediction value of the method. In this way, the reader can better follow the logic of the results.

Reviewer #2: This paper proposed a new approach for miRNA-disease associations prediction. Experimental results indicated that this approach can effectively and efficiently predict miRNA-disease associations. This manuscript is well-written, which makes it easy to understand, although there are some minor issues to be addressed. I have the following specific comments.

1. There are some typos and grammatical errors throughout the paper, the authors should recheck the whole paper carefully to revise these problems.

2. There is an error in Equation (26) and Equation (32) in section of METHODS (D. Optimization algorithm). The A(:,j) should be changed to 〖A(:,j)〗^T. The authors should revise this error..

3. The authors ranked the miRNAs associated with diseases in section of RESULTS (C. Case studies). In my opinion, the authors should specifically introduce how to get the ranking of these miRNAs.

4. Authors compared this model with other MF-based models in section of RESULTS (D. Comparison with MF-based Models). Many matrix factorization-based models have developed for miRNA-disease prediction in a recent time. I think you should compare your model with them.

5. In your paper, the “Gaussian interaction profile kernel similarity” are referred to as “GIP kernel similarity”, but you used “GIP similarity” to referred in a few places in your paper. The authors should pay attention and make changes.

6. Many important computational models for miRNA-disease association prediction published in the top journals such as Bioinformatics and IEEE/ACM Transactions on Computational Biology and Bioinformatics should be discussed and cited. This research field has made much progress in recent several years. Author should mention more recent computational studies for example:

Y. Zhao, X. Chen, and J. Yin, “Adaptive boosting-based computational model for predicting potential miRNA-disease associations,” Bioinformatics, vol. 35, no. 22, pp. 4730-4738, Apr. 2019.

C. Ji, Y. Wang, Z. Gao and L. Li, “A Semi-Supervised Learning Method for MiRNA-Disease Association Prediction Based on Variational Autoencoder,” vol.1, no. 1, p.99, Mar. 2021.

Reviewer #3: In this paper, the authors use similarity network fusion and similarity constrained matrix factorization for miRNA-disease association prediction (SCMFMDA), which completes the missing associated scores between miRNAs and diseases. According to the relevant results, it demonstrates that SCMFMDA is effective for prediction of possible associations of miRNA-disease. However, there are existing some detailed problems.

1. Authors should revise your English writing carefully and eliminate small errors in the paper to make the paper easier to understand.

2. Authors should explain in details the function and impact of parameter ε in Equation (24).

3. The AUC values of global LOOCV of SCMFMDA, MSCHLMDA, GRL-NMF, IMCMDA, ICFMDA and SACMDA were introduced in the section of RESULTS (A. Model validation), but the corresponding values in Fig. 3 look different.

4. There is a reference error in Table III. Authors use ‘a’ and ‘b’ to denote ‘HMDD v3.2 database’ and ‘dbDEMC v2.0 database’ in Table III, respectively, but label ‘H’ and ‘d’ to denote ‘HMDD v3.2 database’ and ‘dbDEMC v2.0 database’ under the Table III.

5. Authors should pay attention to formatting errors which should be further adjusted. For example, there are some fonts are so small in Fig. 2.

**Have the authors made all data and (if applicable) computational code underlying the findings in their manuscript fully available?**

Reviewer #1: **No: **I don't see the link to the source code and dataset they used to train and compare their model with others'

Reviewer #2: Yes

Reviewer #3: Yes

PLOS authors have the option to publish the peer review history of their article (what does this mean?). If published, this will include your full peer review and any attached files.

Reviewer #1: No

Reviewer #2: No

Reviewer #3: No
---

## [Decision Letter · Decision Letter 1]

1 Jun 2021

Dear Mr Li,

Thank you very much for submitting your manuscript "SCMFMDA: Predicting microRNA-disease Associations based on Similarity Constrained Matrix Factorization" for consideration at PLOS Computational Biology. As with all papers reviewed by the journal, your manuscript was reviewed by members of the editorial board and by several independent reviewers. The reviewers appreciated the attention to an important topic. Based on the reviews, we are likely to accept this manuscript for publication, providing that you modify the manuscript according to the review recommendations.

Please revise quickly

Sincerely,

Quan Zou

Guest Editor

PLOS Computational Biology

Ilya Ioshikhes

Deputy Editor

PLOS Computational Biology

[LINK]

Please revise quickly

Reviewer's Responses to Questions

**Comments to the Authors:**

Reviewer #1: The authors have made substantial efforts to address the reviewers' concerns and the revised manuscript improved from many aspects. I have one more question, in the data availability section, the authors claimed "All relevant data are within the manuscript and its Supporting Information files" , yet I haven't found the code and datasets they used in their supporting information files or any link to public website for data sharing in their manuscript. Do I miss it or they don't provide it?

Reviewer #2: All my concerns have been solved.

Reviewer #3: The question of the manuscript has been answered correctly. The paper is acceptable.

**Have the authors made all data and (if applicable) computational code underlying the findings in their manuscript fully available?**

Reviewer #1: **No: **I haven't found the code and datasets they used in their supporting information files or any link to public website for data sharing in their manuscript.

Reviewer #2: Yes

Reviewer #3: Yes

PLOS authors have the option to publish the peer review history of their article (what does this mean?). If published, this will include your full peer review and any attached files.

Reviewer #1: No

Reviewer #2: No

Reviewer #3: No

Figure Files:

Data Requirements:

Reproducibility:

References:

---

## [Editor Report · Decision Letter 2]

8 Jun 2021

Dear Mr Li,

We are pleased to inform you that your manuscript 'SCMFMDA: Predicting microRNA-disease Associations based on Similarity Constrained Matrix Factorization' has been provisionally accepted for publication in PLOS Computational Biology.

Best regards,

Quan Zou

Guest Editor

PLOS Computational Biology

Ilya Ioshikhes

Deputy Editor

PLOS Computational Biology

---

## [Editor Report · Acceptance letter]

5 Jul 2021

PCOMPBIOL-D-21-00629R2 

SCMFMDA: Predicting microRNA-Disease Associations based on Similarity Constrained Matrix Factorization

Dear Dr Li,

I am pleased to inform you that your manuscript has been formally accepted for publication in PLOS Computational Biology. Your manuscript is now with our production department and you will be notified of the publication date in due course.

With kind regards,

Kata Acsay
